# Study on the Nucleation and Growth of Pearlite Colony and Impact Toughness of Eutectoid Steel

**Fei Zhang [1,2], Yonggang Zhao [1], Yuanbiao Tan [1,2], Xuanming Ji [1] and Song Xiang [1,2,*]**

1   College of Materials and Metallurgy, Guizhou University, Guiyang 550025, China; fzhang338@163.com (F.Z.); 18786109880@163.com (Y.Z.); ybtan1@gzu.edu.cn (Y.T.); xmji@gzu.edu.cn (X.J.)
2   Key Laboratory for Mechanical Behavior and Microstructure of Materials of Guizhou Province, Guizhou University, Guiyang 550025, China
*   Correspondence: sxiang@gzu.edu.cn; Tel.: +86-189-8515-1196

**Abstract:** The relationship between microstructure parameters and mechanical properties was studied in this paper. The steel was heat-treated at different austenitizing temperatures to acquire varying microstructure. The results showed that austenite grain size increases with austenitizing temperature, while the pearlite colony size was relatively constant. The strength followed a Hall–Petch relationship with the austenite grain size, but the austenite grain size has nothing to do with the impact toughness. The control unit for determining the impact toughness of pearlitic steel is the pearlite colony size using a comparison method. Further studies have found that, in the hypoeutectoid steel and hypereutectoid steel, the pearlite colony size changes with the austenitizing temperature. However, when the eutectoid steel with a carbon content of 0.81% undergoes the isothermal transformation, the number of grain boundary precipitates is very few. There are many nucleation sites at the grain boundary. The pearlite colonies randomly nucleate at the grain boundaries and grow into the interior of the grains. Simultaneously, new pearlite colonies nucleate by the side of the existing pearlite colony. The intragranular pearlite colonies are also randomly nucleated. These nucleation sites increase the chance of the growing pearlite colonies colliding with each other, eventually resulting in a constant pearlite colony size.

**Keywords:** pearlitic steel; prior austenite grain; pearlite colony; impact toughness; nucleation

## 1. Introduction

Cold-drawn pearlitic steel wires produced by heavy cold drawing are widely used as bridge cables, tire cords, wire ropes, and springs because of their high strength and certain toughness. The pearlitic structure of original pearlitic steel is closely related to the mechanical properties of cold-drawn pearlitic steel wire at a given deformation condition [1]. It is especially critical to control the original multi-layer structure in order to improve the mechanical properties of cold-drawn pearlitic steel wire.

The microstructure of pearlitic steel mainly includes the prior austenite grain, pearlite colony, and interlamellar spacing. Many investigators have carried out extensive research on the relationship between the microstructure and mechanical properties [2–5]. These results showed that refining the microstructure of pearlite steel has become a general point of view to improving its comprehensive mechanical properties. It is generally agreed that the strength follows a Hall–Petch-type relationship with its interlamellar spacing, and refining the pearlite interlamellar spacing results in an increase in strength [6–8]. A pearlitic structure is characterized not only by the interlamellar spacing, but also by the size of colonies and prior austenite grains. Several colonies can nucleate and grow in the bulk of a single initial austenite grain. It was known [9] that the pearlite colony size in the initial austenite grain had a considerable effect on the mechanical properties of pearlitic steel, especially on

its toughness. However, it was known [10] that the toughness was mainly determined by the size of the initial austenite grain, rather than by the size of the pearlite colonies. Liang et al. [11] reported that the size of the austenite grain and pearlite colony increased with the austenitizing temperature, while the impact toughness increased as the prior austenite grain size decreased. Sakamoto et al. [12] found that refining the austenite grain can effectively improve the fracture toughness of medium-high carbon steels, whereas Kavishe et al. [13] reported that the influence of the prior austenite grain size on plane strain fracture toughness is insignificant.

In the above-mentioned studies, the dependences of the mechanical properties of pearlitic steels on the parameters of their structure were determined by traditional methods, namely, impacts, tension, and dynamic bending of the samples. However, there are conflicting results in the literature about the relationship between the toughness and the size of the prior austenite grains and pearlite colonies. There are also few studies on the size of pearlite colonies, and these are short of definite conclusions. Therefore, the primary objective of the present work was to clarify the structure–property relationships. The formation process of the pearlite colony and the control unit of impact toughness were systematically studied in this paper.

## 2. Materials and Methods

In the study, the pearlite steel bar SWRS82B processed by Guizhou Wire Rope Incorporated Corporation (Zunyi, China) is a 70 mm diameter rod, which is obtained by hot rolling and cooled to room temperature in air. The chemical composition is shown in Table 1.

**Table 1.** Chemical composition of SWRS82B in this study (wt%).

| C | Si | Mn | P | Cr | Fe |
|---|----|----|----|----|----|
| 0.810 | 0.180 | 0.840 | 0.014 | 0.272 | Bal. |

In order to prevent the effect of oxidative decarburization on the surface mechanical properties of the sample during heat treatment, the rod of pearlitic steel was first heat treated prior to machining into impact samples. The rod of pearlitic steel was austenitized at various temperatures, ranging from 880 to 1300 °C, for 30 min, and then cooled to 880 °C for 15 min to produce different initial austenite grain sizes. In order to achieve the isothermal transformation of pearlite, the rod of pearlitic steel austenitized at various temperatures was heat treated in a salt bath at 550 °C for 5 min. The schematic diagram in this present work is shown in Figure 1a. Then, charpy U-notch impact samples of $10 \times 10 \times 55$ mm$^3$ in size and tensile samples were cut from the heat-treated rod of pearlitic steel by a wire electric discharge machine. There were three parallel samples in each set of states, and the dimensions of the tensile sample are shown in Figure 1b.

The microstructure was observed using an OLYMPUS (Tokyo, Japan) Gx51 Optical microscope (OM) and ZEISS (Oberkochen, Germany) SUPRA40 field-emission scanning electron microscope (SEM). The room temperature tensile test was carried out at a strain rate of 1 mm/min in a MTS 810 tensile tester (Eden Prairie, MN, America). The room temperature charpy U-notch impact test was carried out on an automatic pendulum impact tester with a pendulum loading speed of 5.3 m/s. After the impact test, the fracture was cut from the impact sample. In order to observe the crack propagation path and measure the pearlite colony size, the fracture was nickel-plated, mechanically polished, and subsequently etched with a 4% nitric acid alcohol solution. Saturated picric acid solution was used to erode austenite grains. The prior austenite grain size and pearlite colony size were determined by a linear intercept method.

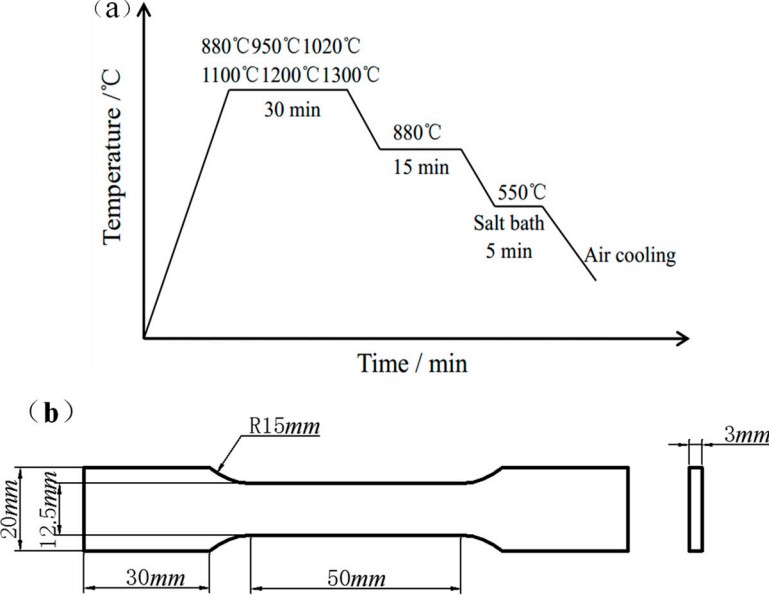

**Figure 1.** (**a**) Process diagram of the heat treatment and (**b**) size of plate structural samples for the tensile test.

## 3. Results

### 3.1. Microstructural Parameters

Figure 2 shows the optical structure of the rod of pearlitic steel after being heat-treated at different austenitizing temperature. It is seen that the austenite grain size increased with the increase in temperature. Figure 3 shows the morphology of the pearlite colony of the rod of pearlitic steel after isothermal pearlite transformation. There was no change in the morphology of the ferrite and Fe$_3$C lamellae in each pearlite colony. The microstructure consists of randomly-oriented pearlite colonies with similar orientations in each colony. It is seen that the pearlite colony size and interlamellar spacing of pearlite hardly change, revealing that the austenitizing temperature has no effect on the pearlite colony size.

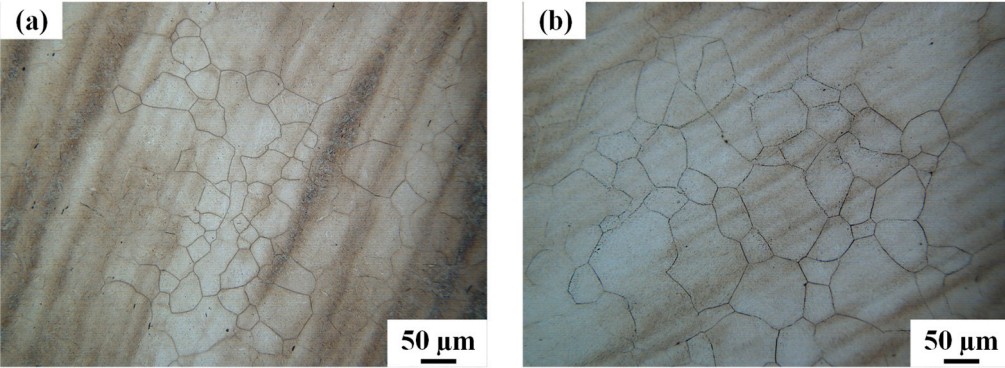

**Figure 2.** *Cont.*

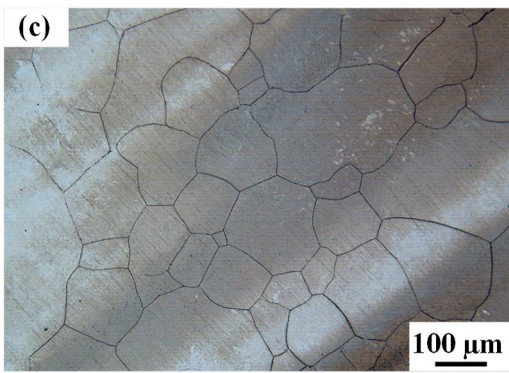

**Figure 2.** Optical micrograph of prior austenite grain at different austenitizing temperatures: (**a**) 880 °C; (**b**) 1020 °C; and (**c**) 1200 °C.

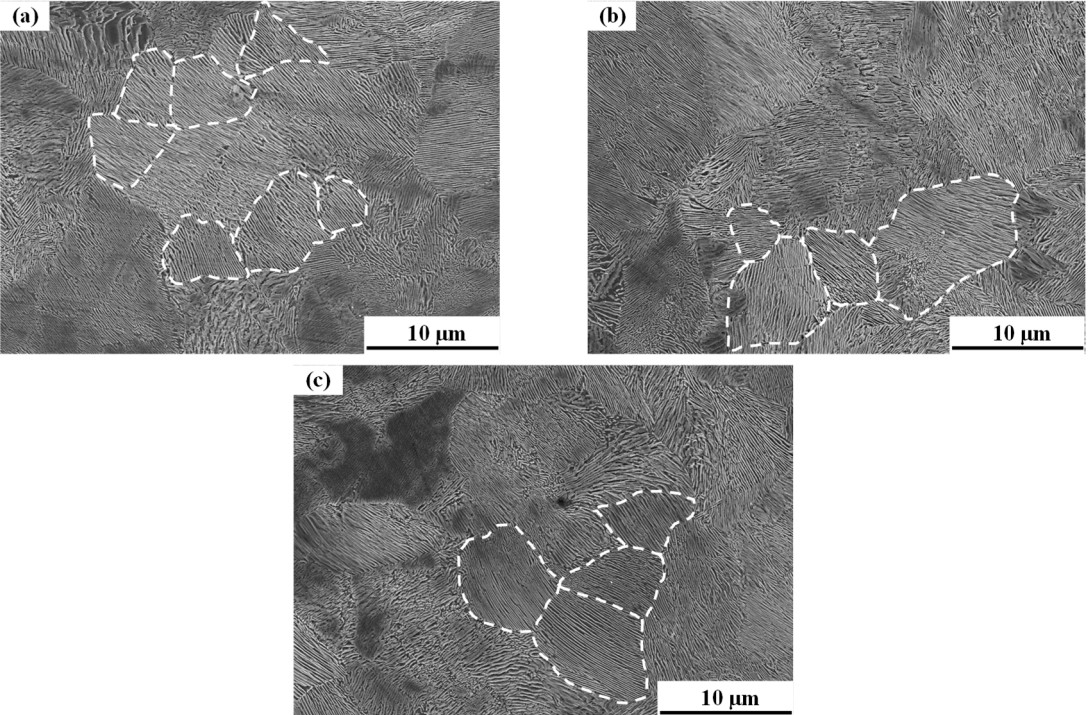

**Figure 3.** Scanning electron microscopy (SEM) micrograph of pearlite structure at different austenitizing temperatures: (**a**) 880 °C; (**b**) 1020 °C; and (**c**) 1200 °C.

### 3.2. Mechanical Properties

Figure 4 shows the engineering stress–engineering strain curves of the rod of pearlitic steel after being heat-treated at different austenitizing temperatures. The tensile strength shows very little variation with the austenitizing temperature over the range from 880 to 1300 °C. It can be observed that the tensile strength of pearlitic steel decreases with the increase of austenitizing temperature, while the yield strength and elongation continuously decrease.

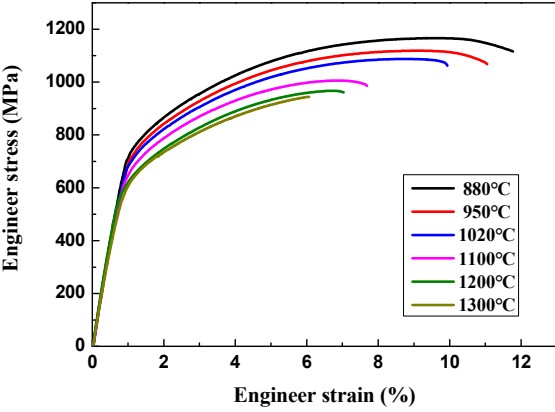

**Figure 4.** Engineer stress–strain curves of different austenitizing temperatures.

The relationship between the microstructure parameters and the impact toughness is listed in Table 2. It is seen that the higher the austenitizing temperature, the larger the austenite grains and the lower the tensile strength. Thus, the variation of strength and elongation may result from the increase of austenite grain size. However, there is no variation observed in the pearlite colony size and impact energy. It is noted that the impact toughness does not change with austenitizing temperature, showing that the austenite grain size is independent of the impact toughness.

**Table 2.** Relationship between microstructure parameters and properties.

| Austenitizing Temperature/°C | Prior Austenite Grain Diameter/μm | Pearlite Colony Diameter/μm | Interlamellar Spacing/μm | Impact Energy/J | Yield Strength/MPa | Tensile Strength/MPa |
|---|---|---|---|---|---|---|
| 880 | 43.39 ± 3 | 6.11 ± 0.4 | 0.1548 | 10 ± 0.5 | 743 | 1154 |
| 950 | 61.29 ± 5 | 5.85 ± 0.5 | 0.1357 | 10 ± 1 | 704 | 1091 |
| 1020 | 80.14 ± 8 | 6.04 ± 0.7 | 0.1310 | 11 ± 1 | 702 | 1086 |
| 1100 | 102.75 ± 10 | 6.10 ± 0.4 | 0.1341 | 10 ± 1 | 650 | 1005 |
| 1200 | 142.48 ± 15 | 5.79 ± 0.5 | 0.1188 | 11 ± 0.5 | 626 | 976 |
| 1300 | 227.26 ± 25 | 5.67 ± 0.4 | 0.1326 | 12 ± 1 | 611 | 959 |

### 3.3. Fracture Morphology

Figure 5 shows the typical fracture morphology of the impact samples austenitized at different temperatures. By inspection and analysis of the fracture morphology, the fracture morphology is similar under the different temperatures. A narrow ductile zone was observed in the impact crack initiation zone and the width of the ductile zone is the same for the impact samples austenitized at different temperatures. In the crack extension zone, the cleavage fracture morphology appears. The cleavage plane size increased with the increasing austenitizing temperature. Figure 6 shows the fracture morphology of tensile samples after austenitizing at different temperatures. It was found that the fracture morphology is obviously different. When the austenitizing temperature is at 880 °C, small amount of dimples and a lot of cleavage fracture with the river pattern can be observed. However, the only cleavage fracture with the river pattern is presented at 1200 °C. This indicates that as the austenitizing temperature increases, the fracture morphology changes from quasi-cleavage to complete cleavage morphology. Although the secondary cracks exist all the time, the number of secondary cracks gradually decreases and the size of the secondary cracks increases with the increase in temperature. Simultaneously, the cleavage facet size increases with the austenitizing temperature. This shows that the larger the prior austenite grains, the larger the cleavage facet. The cleavage facet size is related to prior austenite grain.

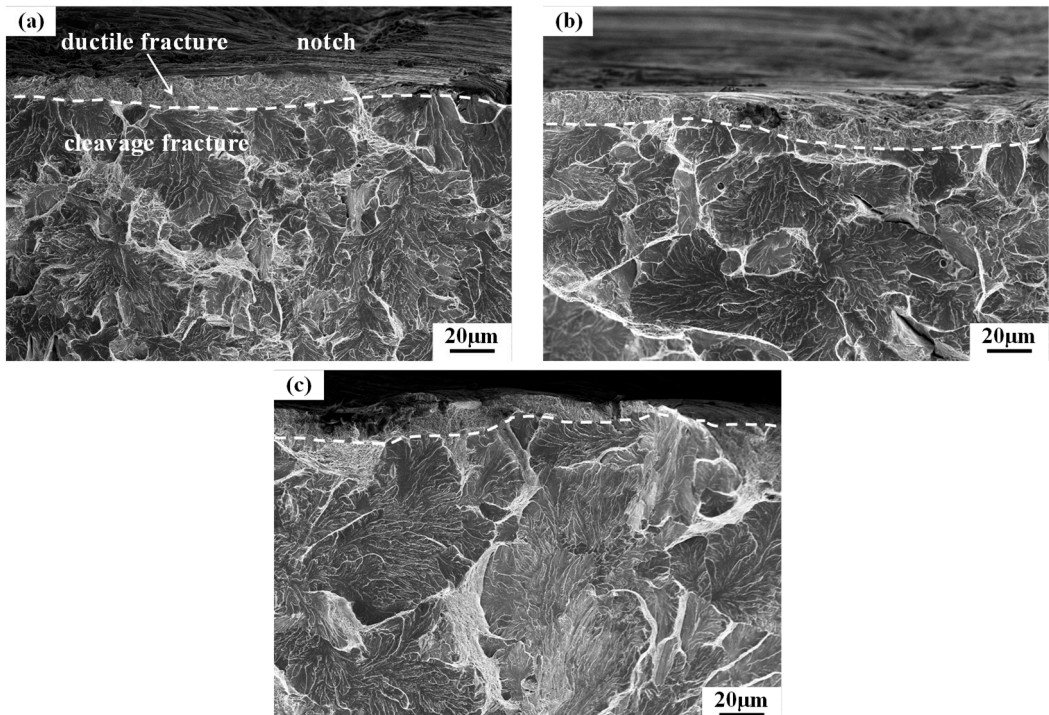

**Figure 5.** Fracture morphology of impact notch at different austenitizing temperatures: (**a**) 880 °C; (**b**) 1020 °C; and (**c**) 1200 °C.

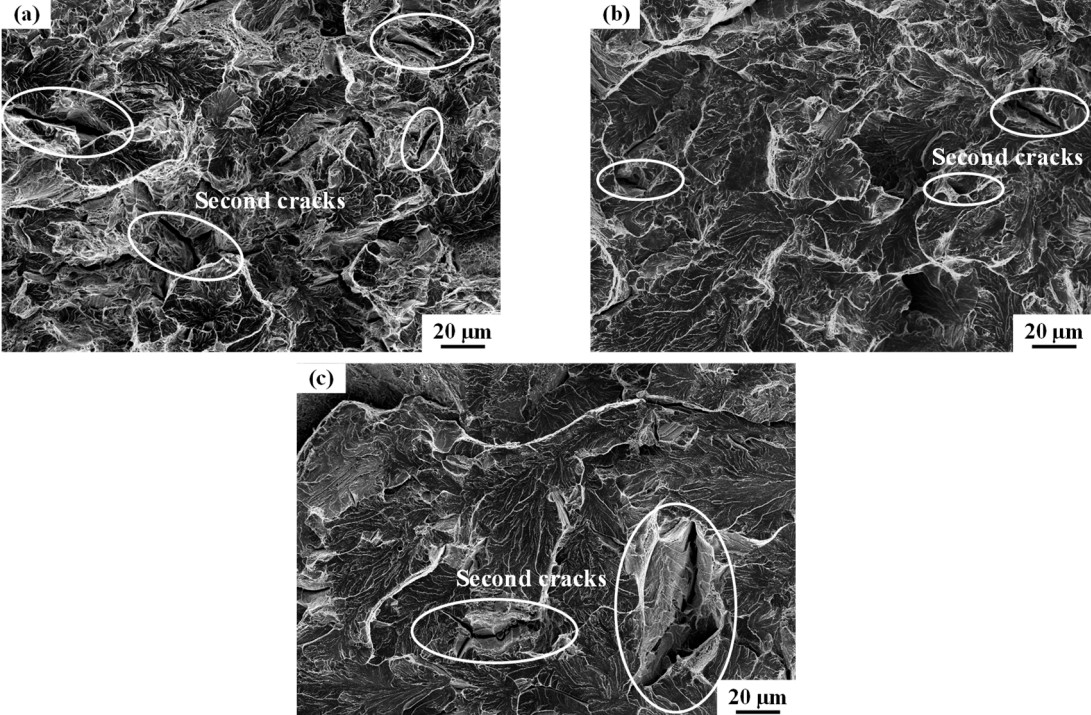

**Figure 6.** Fracture morphology of tensile samples at different austenitizing temperatures: (**a**) 880 °C; (**b**) 1020 °C; and (**c**) 1200 °C.

To further understand the cleavage fracture of the impact samples, the crack propagation path of the impact fracture was analyzed by scanning electron microscopy (SEM). Figure 7 shows the crack propagation path of the impact sample at an austenitizing temperature of 1200 °C. The black area is the nickel layer. It had been found that, in the pearlite colony structure, cleavage cracks most often

propagate along the entire plane, the phase boundary, in a shear direction or perpendicular to the pearlite sheet, and in a changed direction, or stopped, generally at the colony boundaries. The diversity of the crack propagation path may result from the special multi-layer microstructure of pearlite.

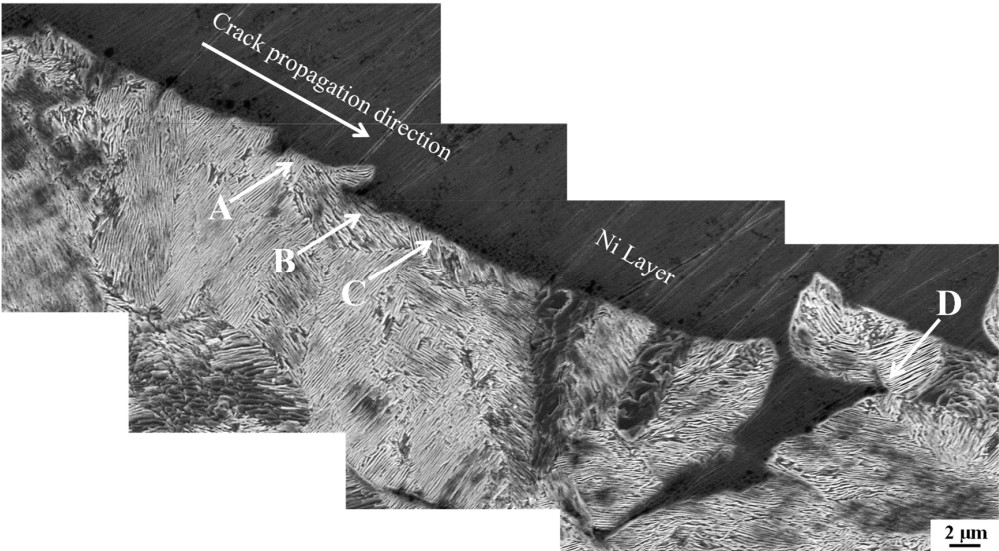

**Figure 7.** Cleavage crack propagation path of the impact sample austenitizied at 1200 °C.

## 4. Discussion

### 4.1. The Nucleation and Growth Mechanism of the Pearlite Colony

The experimental results show that the higher the austenitizing temperature, the larger the austenite grain, while the pearlite colony size is unchanged. Therefore, it is necessary to further analyze the formation mechanism of the pearlite colony. The formation process of the pearlite colony includes nucleation and growth. In the eutectoid steel, it is generally believed that the pearlite preferentially nucleates at the grain boundary. The grain boundaries can act as very effective nucleation catalysts, which can reduce the nucleation energy by a reduction of the interfacial energy. The austenite grain boundary can also enrich carbon and preferentially form cementite. After the cementite nucleus precipitates, the surrounding carbon content decreases and the driving force of ferrite formation increases. By forming ferrite nuclei near the cementite, the process is repeated and the pearlite can grow gradually.

Nucleation is followed by the growth of pearlite colonies. Figure 8 shows the optical microscopy of the microstructure at different moments of the isothermal transformation at 550 °C after austenitizing at 880 °C. When the transformation has proceeded for 5 s, the pearlite begins to nucleate at the grain boundary. After 10 s, some pearlite continues to nucleate at the grain boundary, and the already nucleated pearlite grows into the interior of the grains. After 15 s, a large amount of pearlite is formed. When the transformation has proceeded for 5 s, the transformation is finished, forming an interconnecting network of the pearlite colonies. Figure 9 shows the SEM microscopy of isothermal austenite/pearlite transformation at 550 °C for 10 s after austenitizing at 880 °C. The pearlite colonies nucleate at the grain boundary and then grow into the interior of the grains. Shortly thereafter, the new pearlite colonies nucleate next to the existing pearlite colonies. The pearlite colonies randomly nucleate in the interior of the grains. Figure 10a–h are optical micrographs of 550 °C for 10 s after austenitizing at 880 °C and 550 °C for 15 s after austenitizing at 1200 °C, respectively. However, the higher the austenitizing temperature, the thicker the peeling oxide layer on the surface of the sample after quenching, so the nucleation and growth of pearlite can be observed after 15 s at 1200 °C. The black dotted line indicates an austenite grain. The obvious difference can be seen from the figure. In an austenite grain, the nucleation sites of the grain boundary at 1200 °C are more than

880 °C, and the probability of nucleation in the interior of the grains is larger. According to the theory of solid-state phase transformation, the nucleation of the grain boundary often occurs at the grain boundary when the grain nucleus forms a coherent (or semi-coherent) interface with the grains on one side of the grain boundary, while the non-coherent interface on the other side shows an asymmetric crystal nucleus shape. Also, if the non-coherent nucleation occurs in the interior of the grains, the non-coherent or partially coherent nucleation on the grain boundary always takes precedence over the intragranular generation [14]. As the surface of the pearlite colony is non-coherent with the austenite, the intragranular nucleation lags. In addition, there are a small number of large inclusions in the steel, which can reduce the energy barrier of the nucleation in the interior of the grains [15,16]. The composition of the steel is uniform, and the content of inclusions is the same in the same volume, so the probability of intragranular nucleation occurring is larger in large grains. Offerman and Wilderen et al. [17,18] found that the pearlite colonies gradually evolved from a random distribution at the beginning to a non-random distribution during the formation of complete pearlite, and its results are similar to those of this paper. Simultaneously, Martín et al. [19] also found the existence of intragranular pearlite nucleation under the high-magnification micrographs.

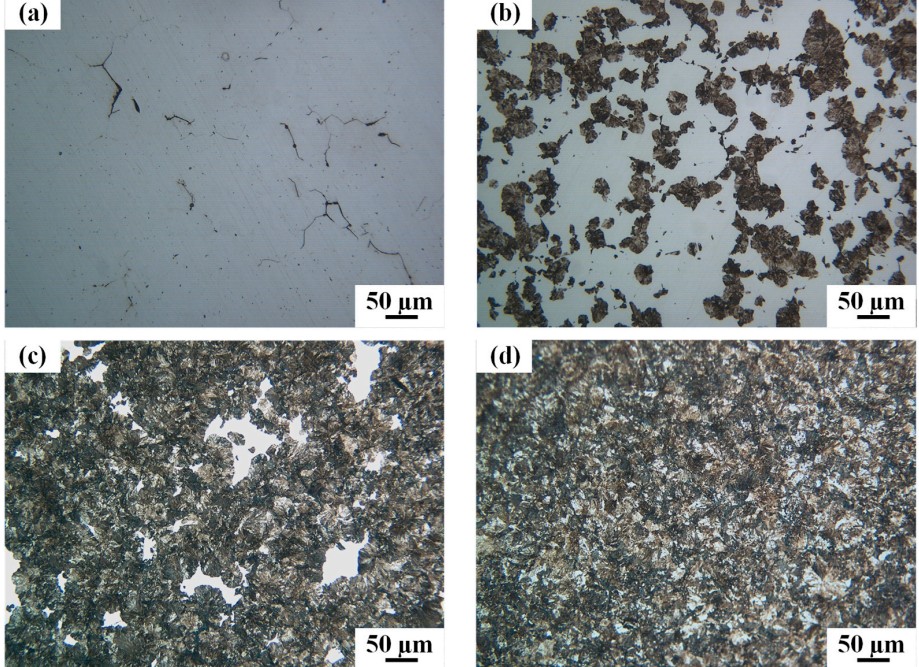

**Figure 8.** Optical microscopy of the microstructure at different moments of the isothermal transformation at T = 550 °C after austenitizing at 880 °C. The samples were quenched to room temperature after 5 s (**a**), 10 s (**b**), 15 s (**c**), and 20 s (**d**).

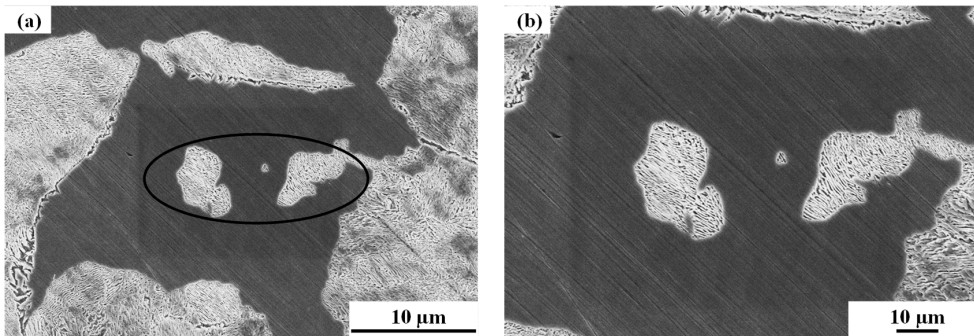

**Figure 9.** SEM microscopy of isothermal austenite/pearlite transformation at 550 °C for 10 s after austenitizing at 880 °C; (**b**) an enlarged view of the black circle in (**a**).

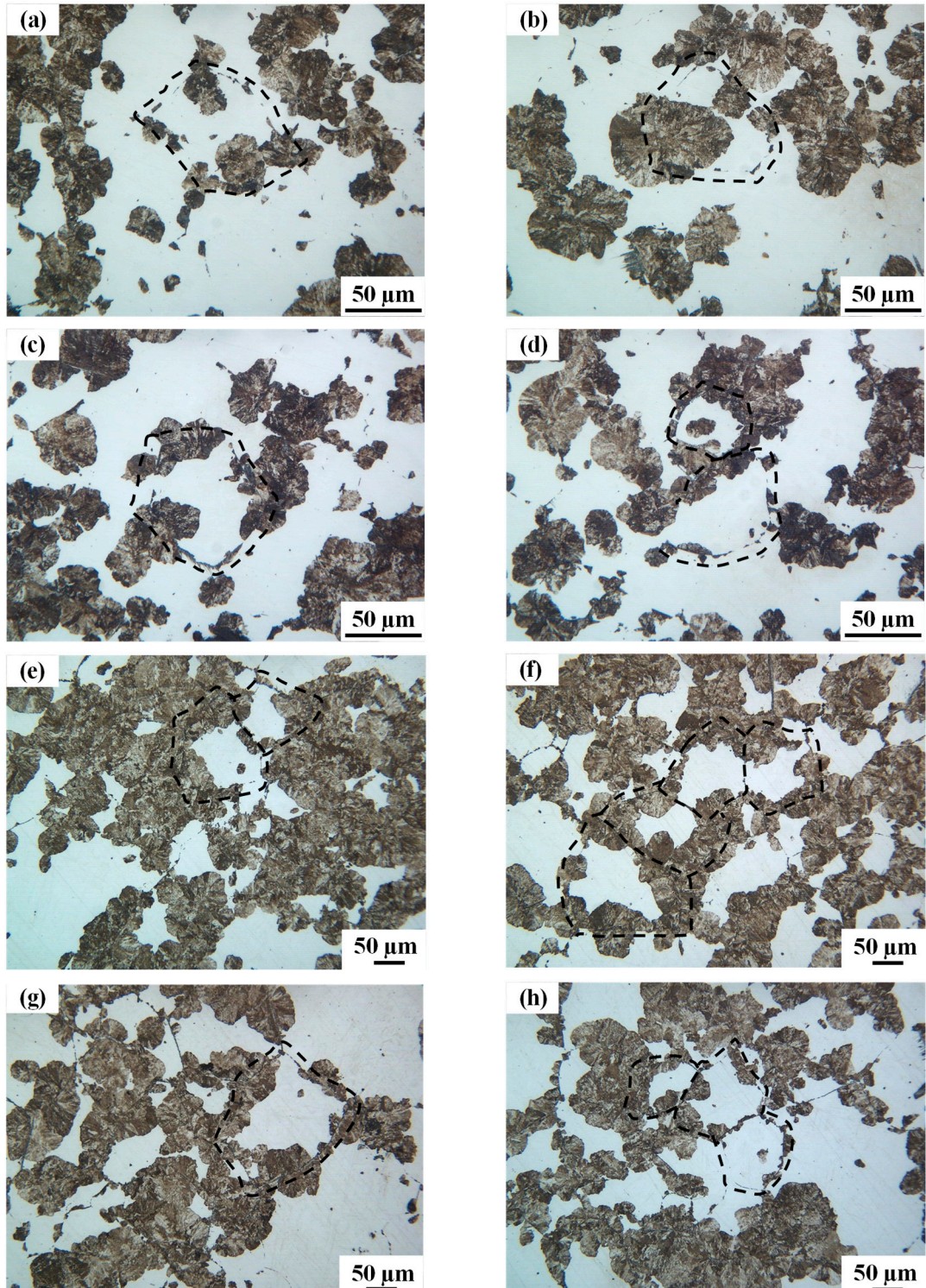

**Figure 10.** (**a–d**) Optical micrographs of 550 °C for 10 s after austenitizing at 880 °C; (**e–h**) Optical micrographs of 550 °C for 15 s after austenitizing at 1200 °C. Black dotted line indicates an austenite grain.

There is a rather interesting result. Previous studies [6,8,11,20] have shown that the pearlite colony size will change with the austenitizing temperature for hypoeutectoid steel and hypereutectoid steel. However, when the near-eutectoid steel with a carbon content of 0.81% in the experimental material undergoes isothermal transformation, the pearlite colony size is relatively constant. Hyzak [10], El-Shabasy [21], and Park et al. [22] also found that the pearlite colony size did not change when

pearlitic steel with a carbon content of 0.81% was austenitized at different temperatures. Therefore, it can be considered that, in pearlitic steel, the pearlite colony size is related to the carbon content, and the number of grain boundary precipitates is affected by the carbon content. Zhang et al. [23] found that the formation of pearlite can be affected by the following different carbon contents: (1) in the hypoeutectoid steel, all the cementite lamellae in pearlite grow from a cementite film, so they have the same orientation; (2) in the hypereutectoid steel, a layer of ferrite film is formed around the eutectoid cementite, and the orientation of the ferrite film is the same as the pearlite ferrite lamellae; and (3) in the eutectoid steel, the pearlite colonies did not grow directly from the austenite grain boundaries. The activated cementite core precipitates along the grain boundary in the form of an unsmooth film or network. Some of the protrusions will grow to form pearlite cementite lamellae. Therefore, the nucleation sites of the eutectoid pearlitic steel are much more than that of the hypoeutectoid steel and the hypereutectoid steel. First, the pearlite colonies randomly nucleate at the grain boundary and grow into the interior of the grains. Then, the new pearlite colonies nucleate next to the existing pearlite colonies. The intragranular pearlite colonies are also randomly nucleated. These nucleation sites increase the chance of the growing pearlite colonies colliding with each other, eventually resulting in a constant pearlite colony size at different austenitizing temperatures for pearlitic steels with a carbon content of 0.81%.

### 4.2. Effect of Microstructure Parameters on Impact Toughness and Strength

The results of the quantitative microscopy are given in Table 2. The observation of each heat-treated structure revealed that the austenite grain size increased with the increase of austenitizing temperature, but the pearlite colony size was relatively stable. The interlamellar spacing of the pearlite depends on the isothermal transition temperature. Because the isothermal transition temperature is constant, the interlamellar spacing is also relatively constant. The interlamellar spacing is the main factor affecting the strength of pearlitic steel. The similar phenomenon has also been reported in the previous research on pearlitic steel with carbon content of 0.81% by Hyzak and Bernstein et al. [10]. The result showed that the austenite grains increased with the increasing austenitizing temperature and time. However, the pearlite colony is not sensitive to various heat treatment parameters such as austenitizing temperature.

It is interesting that as the austenitizing temperature increases, the tensile strength and elongation of pearlitic steel gradually decrease, but the impact energy of the sample is almost unchanged at various austenitizing temperatures. This indicates that the austenite grain size has nothing to do with the impact toughness, whereas it has a lesser relationship with the strength. As there are no means to control both the pearlite colony size and the mechanical properties, there is no way to directly change the pearlite colony size to study its relationship with the impact toughness. Thus, a comparative approach can be adopted. Liang et al. [11] found that the size of austenite grain and pearlite colony increased with the increasing austenitizing temperature, but impact toughness increased with the decreasing austenite grain size. However, the pearlite colony and impact energy are unchanged in the present study, so the pearlite colony size is considered as the control unit of the impact toughness.

Furthermore, Hyzak also stated in the study that the strength may have a lesser relationship with the size of the austenite grain and the pearlite colony. The tensile strength increases by refining these structural variables. Lewandowski et al. [24] believed that the prior austenite grain size has a certain effect on the plasticity of complete pearlitic steel, and the plasticity decreased with the increasing prior austenite grain size. This phenomenon can be explained by the mechanism of fine grain strengthening. According to the Hall–Petch formulas, as shown in Figure 11, the curve of fitting strength and grain diameter is found to be completely in line with the Hall–Petch relationship. The smaller the austenite grain size, the higher the strength. However, the grain size only affects the strength to a lesser extent. The improvement of the strength is explainable by the dislocation accumulation near the grain boundary of the small grain under the stress concentration, so plastic deformation of adjacent grains can occur only under larger applied stress. The improvement of the plasticity is the result of the small strain between the inside of the fine grain and the grain boundary.

The deformation is relatively uniform. There is less chance of cracking due to stress concentration. Thus, it may be subject to large deformation before the fracture. Therefore, the fine austenite grains result in high strength and plasticity at low austenitizing temperatures.

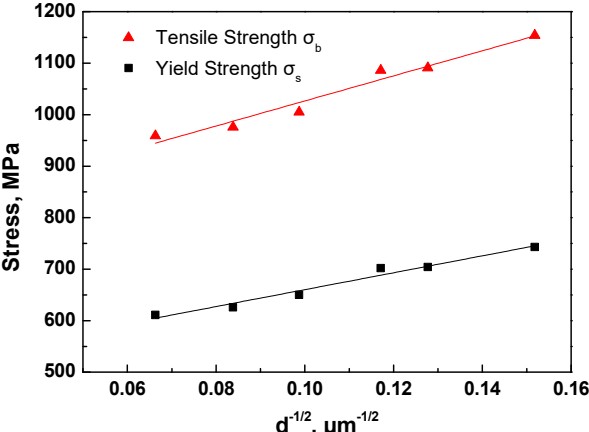

**Figure 11.** Relationship between the measured austenite grains and strength.

### 4.3. Crack Propagation Path

As can be seen from Figure 7, the impact cleavage crack propagation of pearlitic steel is facilitated by the following: (A) being perpendicular to the pearlite sheet, (B) the phase interface between ferrite and cementite, (C) a shear deformation of 45° with the cementite sheet, and (D) cracking along the perlite colony boundary. The letters A, B, C, and D of Figure 7 correlate with the letters A, B, C, and D of the impact cleavage crack propagation. However, previous researchers observed the crack propagation of pearlite by an in situ tensile test, which showed that these four ways basically summarized all of the possible ways of pearlite crack propagation [25–28]. It could be seen that the impact and tensile crack propagation types were same. In addition, the results showed that cleavage cracks most frequently changed direction or stopped at the boundaries between colonies, which may be related to the cleavage fracture plane {100}. Ghosh et al. [29] proposed a new method in which the angle of the {100} cleavage plane of adjacent crystals is considered as the effective grain size. It is believed that, because of the low orientation angle between the {100} cleavage planes of adjacent crystals, the cleavage cracks can propagate along the high angle grain boundaries without any angular deviation. Zhou et al. [30] further confirmed that the cleavage crack propagation is controlled by the pearlite block with an adjacent orientation angle of the pearlite colony greater than 15°, rather than all pearlite colonies.

### 5. Conclusions

In this paper, the control unit for determining the impact toughness of pearlitic steel and the nucleation and growth mechanism of pearlite colony were investigated, and the following conclusions may be obtained:

(1)　The austenite grain size increases with the austenitizing temperature, while the pearlite colony size was relatively constant. The reason for this result is that the pearlitic steel with complete eutectoid structure has more nucleation sites at the grain boundary than the hypoeutectoid steel and hypereutectoid steel. The pearlite colonies randomly nucleate at the grain boundary and grow into the interior of the grains. Simultaneously, the new pearlite colonies nucleate next to the existing pearlite colonies, and the intragranular pearlite colonies are also randomly nucleated. These nucleation sites increase the chance of the growing pearlite colonies colliding with each other, eventually resulting in a constant pearlite colony size.

(2)　The prior austenite grains have a lesser relationship with the strength, conforming to a Hall–Petch-type relationship. The smaller the prior austenite grains, the higher the strength and

the better the plasticity. However, the prior austenite grain size has no impact on the impact toughness. The control unit for determining the impact toughness of pearlitic steel is the pearlite colony size.

(3)  The impact cleavage crack propagation path of pearlitic steel is facilitated by the following: (A) being perpendicular to the pearlite sheet, (B) the phase interface between ferrite and cementite, (C) a shear deformation of 45° with the cementite sheet, and (D) cracking along the perlite colony boundary.

**Author Contributions:** Conceptualization, S.X.; methodology, software, investigation, and writing—review and editing, F.Z.; data curation, Y.Z.; resources, Y.T. and X.J.; visualization, supervision, project administration, and funding acquisition, S.X.

**Funding:** This research was funded by the National Natural Science Foundation of China (Grant Nos. 51661006, 51774103, and 51974097), the Program of "One Hundred Talented People" of Guizhou Province (Grant No. 20164014), Guizhou Province Science and Technology Project (Grant Nos. 20175656, 20175788, 20191414, and 20192163), and the Program for Innovative Research Team of Guizhou Province Education Ministry (Grant No.2016021).

**Conflicts of Interest:** The authors declare no conflict of interest.

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
