# Peer review of "Study on the Nucleation and Growth of Pearlite Colony and Impact Toughness of Eutectoid Steel"

_metals, doi:10.3390/met9111133_

Round 1
Reviewer 1 Report
The authors studied the relationship between the microstructure parameters and mechanical properties on steel SWRS82B. The properties analyzed was impact behavior , tension test and bending .
Point 1: Abstract doesn’t reflect the novelty of the work
Point 2: The bibliography is poor and the authors should describe more clearly the state of the art and the added value if their work in the described contest. Authors could also insert other approach to find a correlation between the thermal treatment and the mechanical properties by using other processes such ad laser for example. The reviewer suggest some literature reference: - Guarino, S., Barletta, M., & Afilal, A. (2017). High Power Diode Laser (HPDL) surface hardening of low carbon steel: Fatigue life improvement analysis. Journal of Manufacturing Processes, 28, 266–271. https://doi.org/10.1016/ j.jmapro.2017.06.015
- Idan, A. F. І., Akimov, O., Golovko, L., Goncharuk, O., & Kostyk, K. (2016). The study of the influence of laser hardening conditions on the change in properties of steels. Eastern-European Journal of Enterprise Technologies, 2(5(80)), 69. https://doi.org/10.15587/1729-4061.2016.65455
- Guarino, S., & Ponticelli, G. S. (2017). High Power Diode Laser (HPDL) for Fatigue Life Improvement of Steel: Numerical Modelling. Metals, 7(10), 447. https://doi.org/10.3390/met7100447
Finally authors should give a mention on the scientific and (eventually) industrial interest.
Point 3: Authors should provide a more accurate description of the device characteristics and of the experimental thermal treatment and characterization procedure
Point 4: How many tests the authors have done to obtain the results? Did the experimentation provide for replications? How many replications?
Point 5: Conclusion should be improved
Author Response
Point 1: Abstract doesn’t reflect the novelty of the work 

Response 1: The novelty of the work has been given in the abstract. The novelty of the work is that the pearlite colony size of the pearlitic steel with a carbon content of 0.81% does not change after heat-treated at different austenitizing temperature and the pearlite colony size is the control unit of the impact toughness.
Point 2: The bibliography is poor and the authors should describe more clearly the state of the art and the added value if their work in the described contest. Authors could also insert other approach to find a correlation between the thermal treatment and the mechanical properties by using other processes such ad laser for example. The reviewer suggest some literature reference: - Guarino, S., Barletta, M., & Afilal, A. (2017). High Power Diode Laser (HPDL) surface hardening of low carbon steel: Fatigue life improvement analysis. Journal of Manufacturing Processes, 28, 266–271. https://doi.org/10.1016/ j.jmapro.2017.06.015
- Idan, A. F. І., Akimov, O., Golovko, L., Goncharuk, O., & Kostyk, K. (2016). The study of the influence of laser hardening conditions on the change in properties of steels. Eastern-European Journal of Enterprise Technologies, 2(5(80)), 69. https://doi.org/10.15587/1729-4061.2016.65455
- Guarino, S., & Ponticelli, G. S. (2017). High Power Diode Laser (HPDL) for Fatigue Life Improvement of Steel: Numerical Modelling. Metals, 7(10), 447. https://doi.org/10.3390/met7100447
Finally authors should give a mention on the scientific and (eventually) industrial interest.
Response 2: Some bibliography is poor, but they are classic literature for pearlitic steel, such as 10. Hyzak, J.M.; Bernstein, I.M. The role of microstructure on the strength and toughness of fully pearlitic steels.Metall. Trans. 1976, 7A, 1217–1224. and 21. Park, Y.J.; Bernstein, I.M. The process of crack initiation and effective grain size for cleavage fracture in pearlitic eutectoid steel. Metall. Trans. 1979, 10A, 1653–1664. and so on. Simultaneously, the state of the art and the added value has been given in the introduction, and they should be enough. For inserting other approach to find a correlation between the thermal treatment and the mechanical properties, I think it is not appropriate in this paper. Because the main purpose of this paper is to obtain different austenite grain size and the same pearlite colony size by heat treatment, then the control unit of impact toughness is discussed in this paper, meanwhile, this is also the novelty of the work.
Point 3: Authors should provide a more accurate description of the device characteristics and of the experimental thermal treatment and characterization procedure.
Response 3: The devices involved in the experiment are simple, a box type resistance furnace and a salt bath furnace containing nitrate. There are many formulas about the nitrate, so I think it is not necessary to take an accurate description. The accurate description of the experimental thermal treatment and characterization procedure has been given in Materials and Methods.
Point 4: How many tests the authors have done to obtain the results? Did the experimentation provide for replications? How many replications?
Response 4: twice tensile tests, Three times impact tests, Four times OM tests of austenite grain size, twice SEM tests of pearlite colony size had been performed.. There were three parallel samples in each test. The experimentation can provide for replications. It can be replicated at least three times.
Point 5: Conclusion should be improved.
Response 5: Conclusion has been improved.

Reviewer 2 Report
Study on the nucleation and growth of pearlite colony and impact toughness of eutectoid steel
This article studies the relationship between microstructure and properties of a fully pearlitic steel subjected to different austenitizing temperatures. Specifically, the authors obtain that the yield strength and tensile strength depend on the prior austenite grain size (following a Hall-Petch type relationship) and the impact toughness of the pearlite colony size. In addition, the fracture surface and the cracking resulting from the tests, and the microstructure in certain moments of the isothermal austenite/pearlite transformation at 550ºC, have been analyzed.
The article deals with a really interesting topic and presents relevant results, but a few aspects must be clarified before publication.
For each austenitizing temperature, how many standard tension tests have been performed? And how many Charpy impact tests have been performed? The linear intercept method is used to determine the prior austenite grain size. What method is used to determine the size of the colonies? Units should be indicated in Figure 1(b). In the header row of Table 2, it should appear prior austenite grain diameter and pearlite colony diameter (instead of prior austenite grain size and pearlite colony size). The value of interlamellar spacing should be added to Table 2. The tensile strength is engineering value or true value in Table 2? It should be true value. It should be indicate that the letters A, B, C and D of Figure 7 correlate with the letters A, B, C and D that appear throughout the text. The section “4.1. The nucleation and growth mechanism of pearlite colony” is confusing. Why doesn't it appear in results? References are missing in relation to the interface coherent (lines 190 to 196) and the hypoeutectoid and hypereutectoid steels (lines 235 to 254). The sentence corresponding to the lines 185 to 187 must be clarified: “However, the higher the austenitizing temperature, the thicker the peeling oxide layer on the surface of the sample after quenching, so the nucleation and growth of pearlite can be observed after 15 s at 1200ºC.” In relation to the previous question, why the micrographs in Figure 10 (a-d and e-h) not correspond to the same transformation time? Although the average values of the prior austenite grain size are 43 and 142 μm, for steel after austenitizing at 880ºC and after austenitizing at 1200ºC (respectively), the difference in size is not observed in the selected prior austenite grains of the micrographs in Figure 10. The selected colonies do not seem very representative of the microstructure. In the line 310, please clarify the sentence: “adjacent pearlite colonies greater than 15º”? It would be convenient a review of the text to fix small errors: For example in line 31 “The mechanical properties of cold-drawn pearlitic steel wire is closely” must be “The mechanical properties of cold-drawn pearlitic steel wire are closely”.Author Response
Point 1: For each austenitizing temperature, how many standard tension tests have been performed? And how many Charpy impact tests have been performed?
Response 1: For each austenitizing temperature, twice standard tension tests and three times Charpy impact tests had been performed.
Point 2:The linear intercept method is used to determine the prior austenite grain size. What method is used to determine the size of the colonies?
Response 2: The linear intercept method was also used to determine the size of the colonies, and it was modified in the paper.
Point 3:Units should be indicated in Figure 1(b). In the header row of Table 2, it should appear prior austenite grain diameter and pearlite colony diameter (instead of prior austenite grain size and pearlite colony size). The value of interlamellar spacing should be added to Table 2. The tensile strength is engineering value or true value in Table 2? It should be true value. It should be indicate that the letters A, B, C and D of Figure 7 correlate with the letters A, B, C and D that appear throughout the text.
Response 3: They had been modified in the paper.
Point 4: The section “4.1. The nucleation and growth mechanism of pearlite colony” is confusing. Why doesn't it appear in results?
Response 4: Because the pearlite colony size is found to be constant with the austenitizing temperature in the experimental results. So the nucleation and growth mechanism of pearlite colony was researched in the discussion part, exploring why the pearlite colony size is unchanged.
Point 5: References are missing in relation to the interface coherent (lines 190 to 196) and the hypoeutectoid and hypereutectoid steels (lines 235 to 254).
Response 5: They had been modified in the paper.
Point 6: The sentence corresponding to the lines 185 to 187 must be clarified: “However, the higher the austenitizing temperature, the thicker the peeling oxide layer on the surface of the sample after quenching, so the nucleation and growth of pearlite can be observed after 15 s at 1200ºC.” In relation to the previous question, why the micrographs in Figure 10 (a-d and e-h) not correspond to the same transformation time?
Response 6: Because in the experimental research, the higher the austenitizing temperature, the thicker the peeling oxide layer on the surface of the sample after quenching to room temperature. In the same transition time, although the thickness of the pearlite transformation on the surface of the sample is the same, the thickness of the peeling oxide layer on the surface of the sample is different at different austenitizing temperatures after quenching to room temperature. Therefore, the nucleation and growth of pearlite can be observed after 10s at 880℃ and after 15s at 1200℃. So the micrographs in Figure 10 (a-d and e-h) not correspond to the same transformation time.
Point 7: Although the average values of the prior austenite grain size are 43 and 142 μm, for steel after austenitizing at 880ºC and after austenitizing at 1200ºC (respectively), the difference in size is not observed in the selected prior austenite grains of the micrographs in Figure 10. The selected colonies do not seem very representative of the microstructure.
Response 7: The difference in size can be observed in the selected prior austenite grains of the micrographs in Figure 10. Although the scale values are the same, the length of the scale is different.
Point 8: In the line 310, please clarify the sentence: “adjacent pearlite colonies greater than 15º”?
Response 8: It should be adjacent orientation angle of pearlite colony greater than 15º. It had been modified in the paper.
Point 9: It would be convenient a review of the text to fix small errors: For example in line 31 “The mechanical properties of cold-drawn pearlitic steel wire is closely” must be “The mechanical properties of cold-drawn pearlitic steel wire are closely”. 

Response 9: They have been modified in the paper.

Round 2
Reviewer 1 Report
The authors made the revisions as requested. The paper can be accepted in present form.